# An Ultra-Rare Manifestation of an X-Linked Recessive Disorder: Duchenne Muscular Dystrophy in a Female Patient

**DOI:** 10.3390/ijms232113076

**Published:** 2022-10-28

**Authors:** Zsuzsanna Szűcs, Éva Pinti, Irén Haltrich, Orsolya Pálné Szén, Tibor Nagy, Endre Barta, Gábor Méhes, László Bidiga, Olga Török, Anikó Ujfalusi, Katalin Koczok, István Balogh

**Affiliations:** 1Division of Clinical Genetics, Department of Laboratory Medicine, Faculty of Medicine, University of Debrecen, 4032 Debrecen, Hungary; 2Doctoral School of Molecular Cell and Immune Biology, University of Debrecen, 4032 Debrecen, Hungary; 32nd Department of Pediatrics, Semmelweis University, 1094 Budapest, Hungary; 4Bioinformatics and Functional Genome Analysis Research Group, Department of Biochemistry and Molecular Biology, Faculty of Medicine, University of Debrecen, 4032 Debrecen, Hungary; 5Department of Genetics and Genomics, Institute of Genetics and Biotechnology, Hungarian University of Agriculture and Life Sciences, 2100 Gödöllő, Hungary; 6Department of Pathology, Faculty of Medicine, University of Debrecen, 4032 Debrecen, Hungary; 7Medical and Health Science Centre, Department of Obstetrics and Gynaecology, Faculty of Medicine, University of Debrecen, 4032 Debrecen, Hungary; 8Department of Human Genetics, Faculty of Medicine, University of Debrecen, 4032 Debrecen, Hungary

**Keywords:** *DMD*, Duchenne muscular dystrophy, NGS, next-generation sequencing, translocation

## Abstract

Duchenne muscular dystrophy (DMD) is the most common inherited muscle dystrophy. Patients are characterized by muscle weakness, gross motor delay, and elevated serum creatinine kinase (CK) levels. The disease is caused by mutations in the *DMD* gene located on the X chromosome. Due to the X-linked recessive inheritance pattern, DMD most commonly affects males, who are generally diagnosed between the age of 3–5 years. Here we present an ultra-rare manifestation of DMD in a female patient. Cytogenetic examination showed that she has a t(X;10)(p21.1;p12.1) translocation, which turned out to affect the *DMD* gene with one of the breakpoints located in exon 54 (detected by genome sequencing). The X-inactivation test revealed skewed X-inactivation (ratio 99:1). Muscle histology and dystrophin immunohistochemistry showed severe dystrophic changes and highly reduced dystrophin expression, respectively. These results, in accordance with the clinical picture and a highly elevated serum CK, led to the diagnosis of DMD. In conclusion, although in very rare cases, DMD can manifest in female patients as well. In this case, a balanced X-autosome reciprocal translocation disrupts the *DMD* gene and skewed X-inactivation leads to the manifestation of the DMD phenotype.

## 1. Introduction

### 1.1. Duchenne Muscular Dystrophy

Duchenne muscular dystrophy (DMD) is an X-linked recessive dystrophin-associated neuromuscular disorder (MIM # 310200) occurring in 15.9 to 19.5 per 100,000 live male births [1].

Generally, affected boys are diagnosed with DMD in early childhood, between ages 3 to 5 years, and present with symptoms like gross motor delay, altered gait, difficulty rising from the ground, frequent falls, and elevated serum creatinine kinase (CK) levels. Muscle weakness presents at first in the proximal lower limbs and trunk, then later in the upper limbs and distal muscles. Progression occurs after about the age of 6 years, and affected children become wheelchair-bound by the age of about 12 years. Joint contractures are common, and scoliosis also develops at some point, impacting the respiratory vital capacity as well. Chronic respiratory insufficiency develops, together with sleep-disordered breathing with or without hypercapnia and hypoventilation. Cardiac symptoms generally present after the age of 10 years in the form of dilated cardiomyopathy and arrhythmias. In addition to intellectual disability and speech or global developmental delay, DMD patients can also present with attention deficit hyperactivity disorder and autism spectrum disorder [2]. If given proper treatment, the majority of the patients can live until the age of 20 to 40 years, when they usually die of cardiac and/or respiratory failure [3].

The disease itself is caused by mutations in the *DMD* gene located on the reverse strand of the X chromosome (Xp21.2-Xp21.1); the canonical transcript (NM_004006.3) contains 79 exons and encodes the 3686 amino acid long dystrophin protein. Frameshift and nonsense mutations in the *DMD* gene disrupt dystrophin function, resulting in connection loss between the actin cytoskeleton and extracellular matrix. In this case, muscle fibers are easily damaged during contraction, which results in chronic muscle damage, inflammation, and eventually loss of muscle function, characteristic for the DMD phenotype [4]. *DMD* gene mutations not disrupting the open reading frame cause a milder phenotype, Becker muscular dystrophy, the ‘reading frame rule’ explaining the different spectra of the disorder [3,5].

Defects in muscle contraction can lead to several types of diseases. The ones occurring specifically in muscle tissue proteins affect their function, thus leading to severe phenotypic presentations, such as the ones mentioned above [6]. The dystrophin protein is part of a large glycoprotein complex in the skeletal muscle sarcolemma and acts as a mechanical link between the extracellular matrix and the cytoskeleton [7]. Dystrophin plays an important role in muscle contraction, and its absence, which is the cause of DMD, leads to a damaged muscle membrane, resulting in an elevated serum CK level. Over time, the muscle progressively loses its contractile function due to cycles of muscle fibre necrosis, degeneration and regeneration, and fatty replacement of the muscle tissue [8].

The full length of the dystrophin protein is expressed in all striated skeletal, smooth and cardiac muscles, whereas shorter isoforms are expressed in brain cells and the retina [9].

### 1.2. DMD Carrier Females

Since DMD is inherited X-linked recessively, it usually only affects hemizygous males. The majority of female carriers are indeed asymptomatic. Up to 20% of them have mild-to-moderate muscle weakness, CK levels are elevated in about 50–60% of the cases, and dilated cardiomyopathy may also be present in about 8% [10]. Even in the absence of skeletal muscle symptoms, females with *DMD* gene variations are estimated to have a lifetime risk of developing cardiomyopathy of up to 17% [11]. Further symptoms that can present include clumsiness in childhood, myalgia/cramps, unexplained abdominal or chest pain, pseudohypertrophy of the calf muscle, and severe gait problems [12]. However, in rare cases, when balanced translocations occur and disrupt the *DMD* gene, heterozygous females can also have the same clinical presentation and disease progression as hemizygous males [8].

### 1.3. Reciprocal Translocations

Translocations are structural chromosome abnormalities that lead to the rearrangement of chromosomes. When a break occurs in two chromosomes and the broken segments exchange, a reciprocal translocation occurs, forming two new derivative chromosomes. When the breakage occurs in females on the X chromosome and the breakpoint disrupts a protein-coding gene, females can also be affected with an X-linked recessive disorder (e.g., DMD). Depending on the size of the exchanged fragments, the reciprocal translocation can be detected either by chromosomal banding studies and/or FISH (Fluorescence In Situ Hybridization) [13].

### 1.4. X Chromosome Inactivation

Generally, one of the two X chromosomes is randomly inactivated during early embryonic development, leaving about 50% of the maternally and a roughly equal percentage of the paternally derived chromosomes active [14]. Chromosomal translocations may result in non-random X chromosome inactivation, which in turn can lead to the expression of a mutant allele, and as a result, female carriers of X-linked recessive disorders will also present with clinical symptoms [15]. The X-inactivation pattern is considered random for values ≤ 80% (ratios ≤ 80:20), moderately skewed for values between 80–90%, and highly skewed for values ≥ 90% [16]. Cases of X-autosomal translocations affecting the *DMD* gene and resulting in preferential inactivation of the normal X chromosome have been described in the literature before [17,18].

### 1.5. Translocations in Females with DMD Phenotype—Previously Reported Cases in the Literature

To date, 25 balanced reciprocal translocation cases involving the *DMD* gene and causing DMD phenotype in females have been reported in the literature. In Table 1, cases from the HGMD Professional (Human Gene Mutation Database Professional 2022.3), the Leiden Muscular Dystrophy pages, and PubMed are summarized.

Here, we present an ultra-rare manifestation of DMD in a female patient caused by an X-autosomal translocation disrupting the *DMD* gene.

## 2. Results

### 2.1. Case Presentation

The index female patient was born by caesarean section at the 41st week of gestation as the first child of healthy parents. She required temporary respiratory support after birth and received parenteral antibiotic treatment for 5 days. Speech development was normal, but motor development was delayed. Unsteady gait and imbalance were described at the age of 3 years, and later poor coordination at age 6. In spite of physical and occupational therapy, no improvement of symptoms was observed, therefore, a neuromuscular disease was suspected. Laboratory studies showed highly elevated serum CK (9892 U/L at age 7, 10,694 U/L at age 9 (reference range: 24–195 U/L)). At the age of 9 years, cardiological examination was normal. EMG/ENG (Electromyography/Electroneurography) studies showed a dominant neurogenic pattern with myogenic lesions. There was no family history of neuromuscular disorders. At age 13, gradual deterioration of muscle strength was observed with hypotrophic muscles. Serum CK was 1868 U/L.

### 2.2. Genetic Testing and Muscle Biopsy

Cytogenetic testing was performed at the age of 7 years and revealed a balanced reciprocal translocation, 46,XX,t(X;10)(p21;p12) (Figure 1). Cytogenetic testing of the parents showed that they are not carriers of the translocation, proving that it was de novo.

Muscle biopsy showed severe dystrophic features such as a high degree of variation in fiber size with a mixed appearance of hypertrophic and atrophic fibers, increased internal nucleation (~10%), and numerous basophilic regenerative and degenerative fibers. Necrotic fibers, myophagocytosis, a high degree of pericellular fibrosis (Figure 2A), and adipose tissue infiltrates were also present. NADH (Nicotinamide adenine dinucleotide) staining showed irregular myofibrillar structure in several fibers (Figure 2F). The spectrin expression was reduced in several fibres (Figure 2B). The α, β, γ, and δ sarcoglycans and merosin immunohistochemistry showed normal linear sarcolemmal expression. Dystrophin expression with three anti-dystrophin antibodies directed against the central/core, C-terminal, and N-terminal regions of dystrophin showed segmentally severely reduced or completely lost sarcolemmal staining with the co-presence of fascicles, mainly consisting of normal or hypertrophic fibres, showing normal, linear sarcolemmal dystrophin expression (Figure 2C–E). Negative controls for immunohistochemical reactions are shown in Figure 2G–J. The dystrophin immunohistochemistry has been repeated to confirm the findings.

MLPA (Multiplex Ligation Dependent Probe Amplification) performed at the age of 8 years showed no copy number variations (duplications or deletions) in the *DMD* gene. Further molecular genetic testing at age 9 for polyneuropathy, FSHD (facioscapulohumeral muscular dystrophy), and LGMD (limb-girdle muscular dystrophy) was negative (targeted testing of the *CAPN3, FKRP* and *PMP22* genes was performed).

Array-CGH (comparative genomic hybridization) revealed two copy number variants (CNV) of unknown significance (VUS) (arr[GRCh37] 7q31.1(107692546_107849992)x3 and Xp22.11(22000870_22167026)x3). Array-CGH testing of the parents revealed that one of the CNVs was inherited from the mother (arr[GRCh37] Xp22.11(22,000,870_22,167,026)x3), whereas the other CNV from the father (arr[GRCh37] 7q31.1(107692546_107849992)x3), suggesting that these CNVs were not the cause of the disease.

At age 13, targeted next-generation sequencing (NGS) of the *DMD* gene showed no pathogenic mutations. Whole exome sequencing (WES) was performed subsequently and revealed a pathogenic missense mutation in the *CAPN3* gene, (NM_000070.2) c.2243G>A, p.(Arg748Gln). *CAPN3* is associated with autosomal dominant type 4 (MIM # 618129) and autosomal recessive type 1 (MIM # 253600) LGMDs. The ClinVar database (ID: 128570) reports this variant to be associated with autosomal recessive type 2A LGMD, and it has been previously described in a homozygous or compound heterozygous form, suggesting that our patient, if affected with LGMD, should have a second pathogenic alteration in the *CAPN3* gene as well. MLPA testing revealed no copy number variation in the *CAPN3* gene.

As a secondary finding in WES data, a heterozygous pathogenic frameshift mutation was detected in the *BRCA2* gene (NM_000059.3:c.9097dupA, p.(Thr3033Asnfs*11)). Targeted Sanger sequencing of the parents revealed that it is also present in the mother.

Afterwards, whole genome sequencing (WGS) was performed, and the raw data was used for breakpoint analysis. It confirmed the t(X;10)(p21.1;p12.1) balanced reciprocal chromosomal translocation with the following breakpoints: chrX: 31,676,155—chr10: 26,236,351 and chrX: 31,676,158—chr10: 26,236,355. This result suggests that there is a 4bp deletion on chromosome X and a 3bp deletion on chromosome 10. The breakpoint on chromosome X is located in Xp21.1, in exon 54 of the *DMD* gene, whereas the breakpoint on chromosome 10 is located in 10p12.1, in intron 2 of the *MYO3A* gene.

As a consequence of the translocation, two derivative (der) chromosomes are formed: der(X) and der(10). Chromosome der(X) consists of 10p15.3-p12.1—Xp21.1-Xq, with the following coordinates: chr10:1–26,236,351 (chr10:26,223,196–26,236,351 encodes exon 1 and part of intron 2 of the *MYO3A* gene) and chrX:31,676,158–155,270,560 (chrX:31,676,158–33,229,636 encodes exons 54–1 of the *DMD* gene—containing the first part of both genes). Chromosome der(10) consists of Xp22.33-p21.1—10p12.1–10q with the following coordinates: chrX:1–31,676,155 (chrX:31,137,345–31,676,155 encodes exons 79–54 of the *DMD* gene on the reverse strand, therefore, the first part of the gene is translocated) and chr10:26,236,355–135,534,747 (chr10:26,236,355—26,501,456 encodes the *MYO3A* gene from intron 2 to exon 35 on the forward strand, the first part of the gene is also translocated) (Table 2, Figure 3).

The presence of split reads aligned to the breakpoints can be seen in Figure 4. Part A shows the reads from WGS data. The first part of the sequence of these reads is complementary with the sequence of the *DMD* gene on chromosome X, whereas the other part of the sequence can be aligned to the *MYO3A* gene on chromosome 10. After detecting the split reads in the WGS data, we retrospectively checked the position of the breakpoint in the targeted *DMD* sequencing data as well, where the coverage is higher and indeed, the presence of the split reads could also be detected in a higher number, as seen in part B of Figure 4.

Targeted Sanger sequencing confirmed the presence of the breakpoints, also clarifying that there is a 3bp (TGC) deletion on chromosome 10, as previously assumed based on the bioinformatical analysis of the WGS data. On the other hand, there is no deletion on chromosome X but a 4bp (GGTA) insertion instead (Figure 5 and Figure 6).

The X-inactivation test showed highly skewed X inactivation with a ratio of 99:1, meaning the preferential inactivation of one of the X chromosomes in 99%.

## 3. Discussion

Here, we present the case of a girl with muscular dystrophy of unclear origin and her diagnostic odyssey. She presented with delayed motor development, unsteady gait, imbalance and poor coordination, and laboratory studies showing highly elevated serum CK.

Cytogenetic testing revealed a de novo balanced reciprocal translocation, 46,XX,t(X;10)(p21;p12), which was then followed by a series of molecular genetic tests until the clinical diagnosis could be established. Although the presence of the translocation was detected early in the investigation, DMD was not suspected as it is generally associated with males, not females, and the involvement of the *DMD* gene was not confirmed either at first.

Because of the X-autosome translocation and suspected myopathy, a muscle biopsy was performed, at that time preceding genetic testing of neuromuscular disorders. Muscle biopsy showed severe dystrophic changes. Dystrophin immunohistochemistry with all three antibodies, directed against epitopes of the central/core, C-terminal, and N-terminal regions of dystrophin, showed segmentally severely reduced or completely lost sarcolemmal staining with the co-presence of fascicles showing normal, linear sarcolemmal dystrophin expression. The latter finding might be the result of the presence of an active normal X chromosome, which was shown to be 1% in the peripheral blood but might be different in the muscle.

The WES result was misleading at first as a heterozygous pathogenic missense mutation in the *CAPN3* gene was detected. *CAPN3*-caused calpainopathy and DMD have overlapping clinical features, therefore, this could have been the cause of the disease as well. However, only a single, recessively acting mutation was detected, and the presence of a copy number variant was ruled out by MLPA.

Furthermore, WES revealed a secondary finding in the *BRCA2* gene. *BRCA2* is associated with hereditary breast and ovarian cancer, and it is characterized by an increased risk for breast cancer, ovarian cancer, and to a lesser extent, other cancers (prostate cancer, pancreatic cancer, melanoma) [45]. The cumulative risk of developing breast cancer is 69% and developing ovarian cancer is 17% for *BRCA2* mutation carriers [46]. Having the informed consent of the parents requesting information about the potential incidental findings as well, targeted testing proved the mother to also harbour this mutation.

Breakpoint detection performed from WGS data revealed that the translocation disrupts the *DMD* gene on one of the X chromosomes. Since the breakpoint is located in exon 54, it will either lead to a truncated protein or nonsense-mediated decay of the mRNA. X-inactivation testing showed highly skewed inactivation. The test cannot specify whether the normal or the translocated X is inactivated, but since the patient presents with the clinical picture of DMD, it is most likely that the normal X chromosome becomes inactivated in this case. In addition to this, if the der(X) had become inactive, a large part of chromosome 10 would have been inactivated as well, which would probably be incompatible with life [16].

The other breakpoint is localised in the intron of the *MYO3A* gene. *MYO3A* encodes an actin-dependent motor protein and it is associated with autosomal recessive hearing impairment and deafness [47]. 25 disease-causing mutations have been described in the HGMD Professional 2022.3 and 26 pathogenic or likely pathogenic variants in the ClinVar (19 October 2022), the majority being loss-of function ones. However, in the gnomAD population database, there are roughly 200 alleles with loss-of-function mutations, these being present in the healthy population in heterozygous form. Moreover, our patient does not have any symptoms of hearing impairment, therefore, it is highly unlikely that the disruption caused by the breakpoint causes any symptoms in this case.

Performing WGS in the first place would have significantly reduced the time required to establish a definitive diagnosis. In this case, the first cytogenetic examination was performed when the index patient was 7 years old, but she received the definitive molecular genetic diagnosis confirming DMD only at age 14.

In conclusion, all data, including the clinical picture, the highly elevated serum CK, and the X-autosome translocation disrupting the *DMD* gene with highly skewed X-inactivation, lead to the diagnosis of DMD. In very rare cases, DMD can manifest in female patients as well, especially when a balanced X-autosome reciprocal translocation disrupting the *DMD* gene is accompanied by skewed X-inactivation. To date, 25 DMD translocation cases have been published in the literature in females with DMD phenotype, and here we have presented another case of a girl having t(X;10)(p21.1;p12.1) translocation.

## 4. Methods and Materials

### 4.1. Methods

#### 4.1.1. Molecular Genetic Testing

Cytogenetic evaluation (Giemsa staining and FISH) was performed, whole-chromosome painting probes WCP X, WCP 10, 10p14 (BRUNOL), Kallmann and SHOX (Cytocell) probes were used for FISH analysis.

Genomic DNA was isolated from peripheral blood leukocytes using the QIAamp Blood Mini kit (Qiagen GmbH, Hilden, Germany).

MLPA was performed using SALSA MLPA Probemix P176 CAPN3 for LGMD 2A (MRC Holland, Amsterdam, Netherlands) and SALSA MLPA Probemix P034 DMD-1 for DMD; Becker muscular dystrophy (BMD) according to the manufacturer’s protocol.

Array-CGH was performed at qGenomics (Spain). The analysis was performed using qChip 180K and on Agilent Technologies Platform. Obtained data was analysed using the Genomic Workbench 7.0 software.

Targeted *DMD* gene sequencing was performed on Illumina NextSeq 550 (Illumina, San Diego, CA) sequencer system in 2 × 150 cycle paired-end mode. A custom-made and enrichment-based DNA library preparation kit (Twist Bioscience, South San Francisco, CA) containing the *DMD* gene as well was used.

WES was performed on the Illumina NextSeq 500 sequencer system in 2 × 150 cycle paired-end mode. The Nextera DNA Exome kit (Illumina) was used for library preparation.

WGS was performed on the DNBSEQ-G400 sequencer system in paired-end mode. The MGIEasy Universal DNA Library Prep kit (MGI Tech Co., Ltd., Shenzhen) was used for library preparation.

Sanger confirmation of the breakpoints detected by genome sequencing was performed using the BigDye Terminator v3.1 Cycle Sequencing kit (Applied Biosystems, Foster City, CA, USA) according to the manufacturer’s protocol, with primers designed specifically for the breakpoints by Primer3 (v. 0.4.0): derX-F: 5′- cacatgttccgtctatctctattctc -3′, derX-R: 5′- aaaactgacattcattctctttctca -3′ and der10-F: 5′- ctcccaagctccagtttagc -3′, der10-R: 5′- atctttgccaggcgcagt -3′.

X-inactivation test (AR locus, 21D4591) was performed at Amsterdam UMC (the Netherlands).

#### 4.1.2. Muscle Histology and Dystrophin Immunohistochemistry

Frozen sections of 7 micrometer thickness from muscle biopsy specimens were processed for histology, histochemistry, and immunohistochemistry according to standard protocols [48].

Regarding the detection of sarcolemmal and sarcolemma-associated proteins, immunohistochemistry was performed by using antibodies targeting different dystrophin epitopes: NCL-DYS1 (clone: DY4/6D3), NCL-DYS2 (clone: DY8/6C5) and NCL-DYS3 (clone: DY10/12B2), corresponding to central/core, C-terminal and N-terminal regions, respectively (Novocastra, Newcastle, UK, primary antibody dilution was 1:20 in all cases). The following antibodies were used targeting sarcoglycan alpha, beta, gamma, and delta: NCL-L-a-SARC, clone AD1/20A6; NCL-L-b-SARC, clone BEATASARC1/5B1; NCL-g-SARC, clone 35DAG/21B5; NCL-d-SARC, clone DELTASARC/12C1; (Novocastra, 1:50, 1:100, 1:100, and 1:50, respectively); merosin: NCL-MEROSIN, clone: MER3/22B2 (Novocastra, 1:100) and spectrin NCL-SPEC1, clone RBC2/3D5 (Novocastra, 1:100). Primary antibodies were incubated on slides for 1 h at room temperature.

As negative control, a muscle biopsy sample from an ALS (Amyotrophic Lateral Sclerosis) patient was used. We have succesfully applied the settings described above to detect dystrophinopathy recently [49].

#### 4.1.3. Bioinformatical Analysis, Determination of Translocation Breakpoints

In the case of next generation sequencing, raw data were aligned to the GRCh37 reference genome using the NextGENe software (SoftGenetics, State College, PA, USA); the variant list was uploaded to the Franklin Analysis Platform (Genoox) [50] for variant classification, and HPO terms were used to help with variant prioritization. The gnomAD population database, ClinVar (v 2021-10-10) and HGMD Professional 2020.3 mutation databases were also used in variant prioritization and interpretation.

WGS data were analysed to determine the breakpoints of the translocation. The human reference genome used during the analysis was version GRCh37.

The BWA-MEM (Burrows-Wheeler Aligner, version: v0.7.17) [51] software package was used for mapping low divergent sequences. The SAMtools package (v1.6) [52] was used to sort and index the BAM (binary alignment map) file and then the GATK (Genome Analysis Toolkit) package (v4.1.8.1) [53] was used to filter the duplicates. DELLY (v0.8.7) [54] was used to call only inter-chromosomal translocations (BND—breakend). For manipulation of bcf (binary variant call format) and vcf (variant call format) files, the BCFtools (v1.9) [52] was used, whereas filtering Pass and Precise results from the vcf files (file format is VCF v4.2) was performed with bash commands.

Quality filters were set by DELLY as follows: PASS: PE/SR (paired-end/split read) support = 3 or more, and mapping quality > 20. An additional quality filter that was taken into consideration was that the filtered poor-quality reads or paired-ends and split reads of insufficient number were eliminated from the processing. In the next filtering step, the following values were set: PEs and SRs with median mapping quality (MAPQ (median mapping quality of paired-read) = 60; SRMAPQ = 60), consensus alignment quality of SRs (SRQ) > 0.9 and matching second (reciprocal) translocations with 3′ to 3′ and 5′ to 5′; or 3′ to 5′ and 5′ to 3′ matching connection types.

The remaining translocations after filtering required further classification into false and possible reciprocal translocations by visualizing the breakpoints in Integrative Genomics Viewer (IGV v2.8.12). In addition, gene(s) involved in the translocations were also checked in ENSEMBL (release 104).

## Figures and Tables

**Figure 1 ijms-23-13076-f001:**
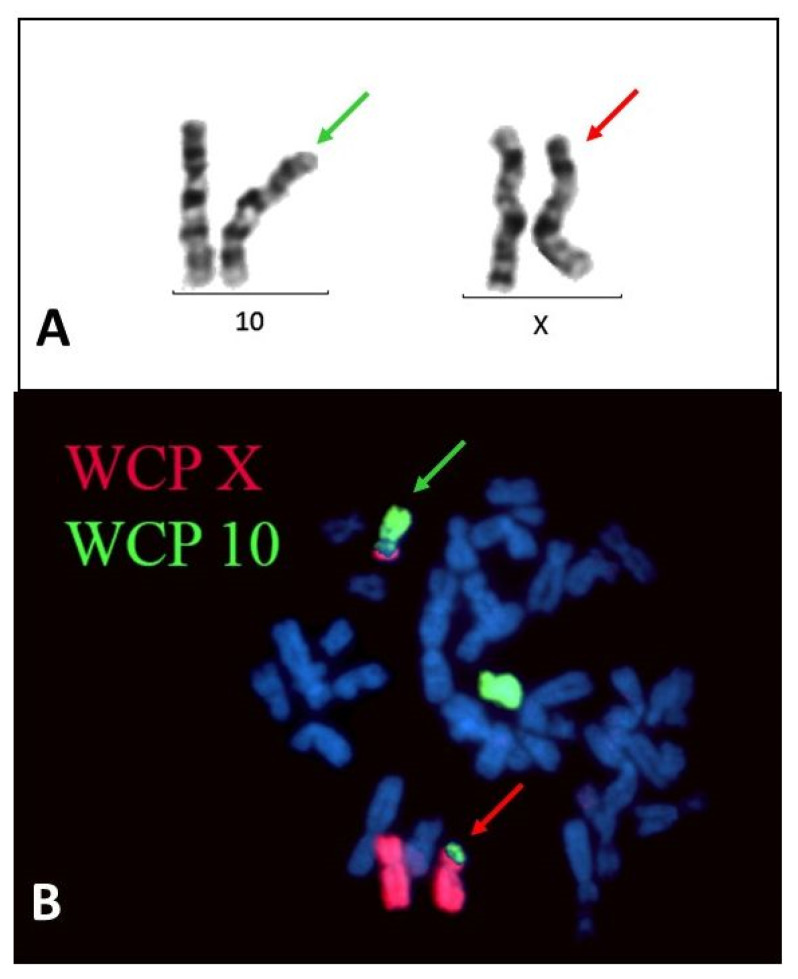
G-banding (**A**) and FISH pattern (**B**) of the t(X;10)(p21.1;p12.1) translocation. In the FISH image, the X chromosomes are visualized by red and chromosome 10 by green whole chromosome painting probes (WCP). The red arrows indicate the derivative X, whereas the green arrows the derivative 10 chromosomes.

**Figure 2 ijms-23-13076-f002:**
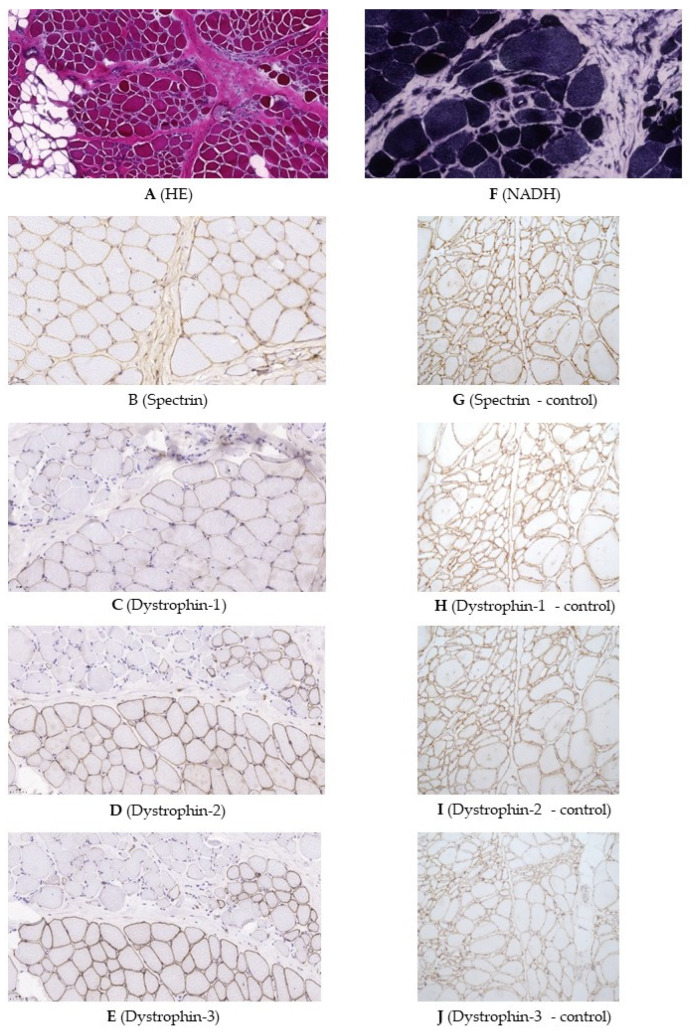
Muscle biopsy, magnifications used between 110×–320×. (**A**) Histological examination showed variation in fiber size, increased number of internal nuclei, necrotic fibers, pericellular fibrosis, and adipose tissue infiltrates (HE: hematoxylin-eosin). (**B**) The spectrin expression was reduced in several fibres. (**C**–**E**) Dystrophin expression showed segmentally severely reduced or completely lost sarcolemmal staining with the co-presence of fascicles showing normal, linear sarcolemmal dystrophin expression. (**F**) NADH-TR (nicotinamide adenine dinucleotide-tetrazolium reductase) staining showed irregular myofibrillar structure in several fibers. (**G**) Spectrin expression in the control. (**H**–**J**) Dystrophin expression in the control.

**Figure 3 ijms-23-13076-f003:**
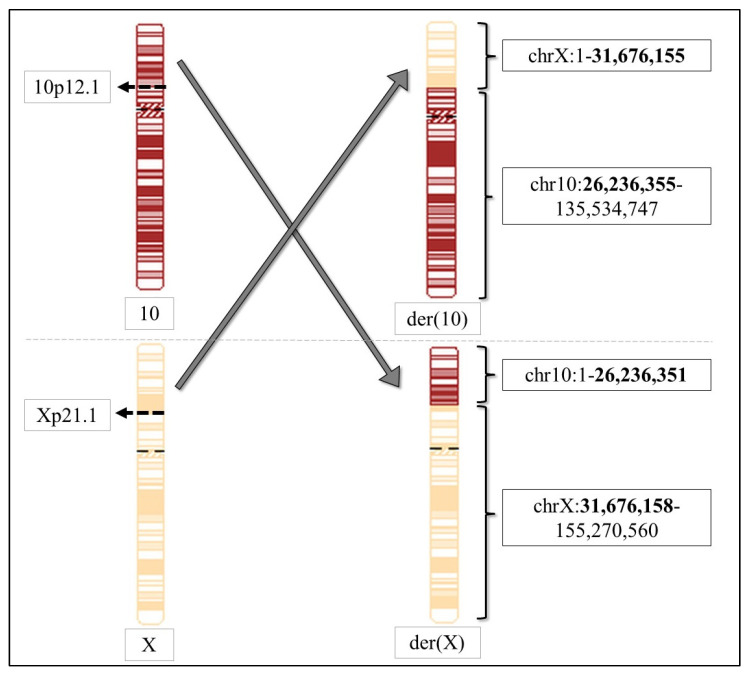
Schematic illustration of the t(X;10)(p21.1;p12.1) balanced reciprocal chromosomal translocation. Breakpoints are shown in bold.

**Figure 4 ijms-23-13076-f004:**
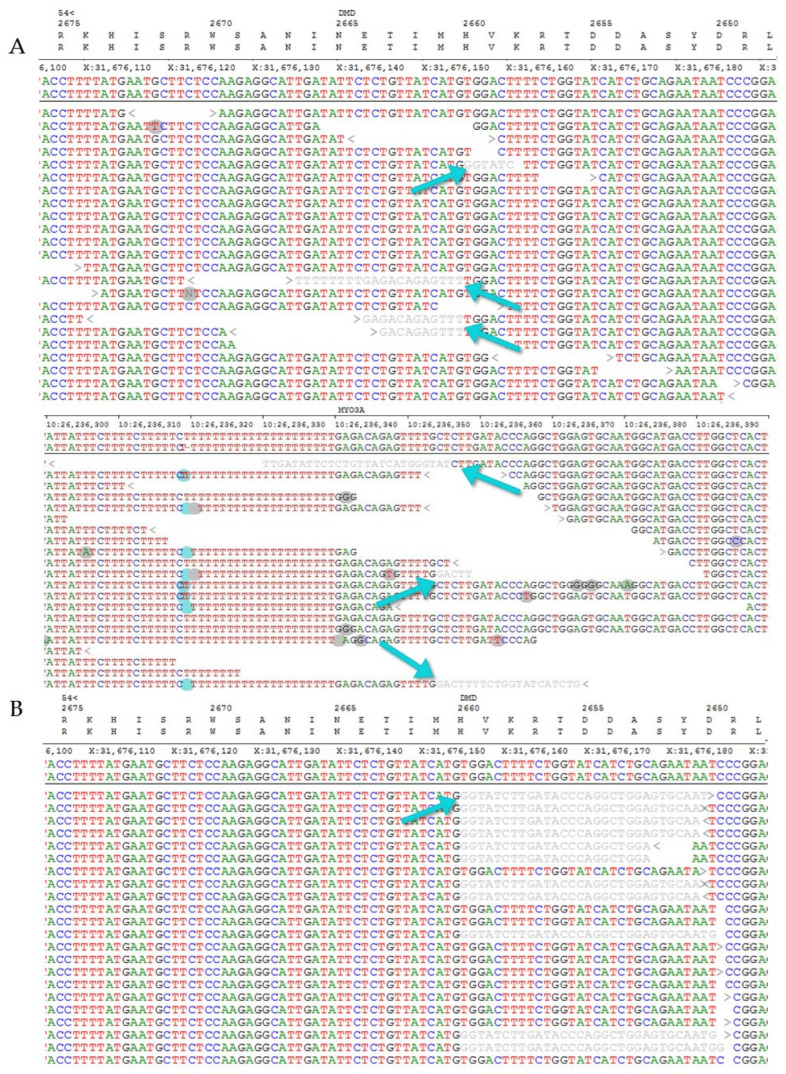
(**A**) The presence of split reads in whole genome sequencing data aligned to the breakpoints (shown in grey, indicated by blue arrows). The first part of the sequence of the reads is complementary with the sequence of the *DMD* gene on chromosome X, as shown in the upper part of the figure, whereas the other part of the sequence can be aligned to the *MYO3A* gene on chromosome 10, as shown in the lower part of the picture. The blue and grey circles in the reads indicate the alignment errors by the software. (**B**) The presence of split reads in the targeted next-generation sequencing data aligned to the breakpoints (shown in grey, indicated by blue arrows) in the *DMD* gene.

**Figure 5 ijms-23-13076-f005:**
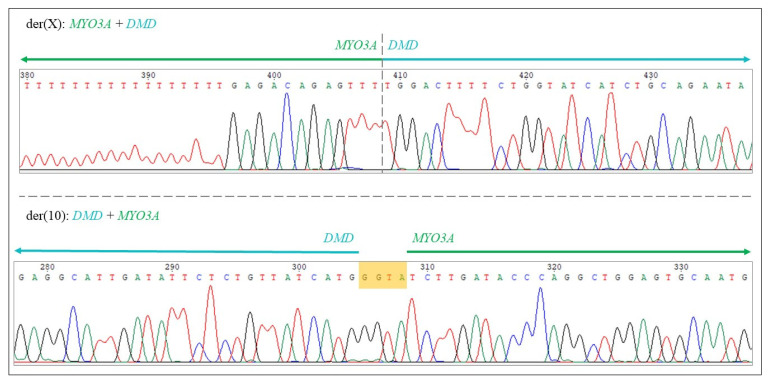
Sanger sequencing data showing the breakpoints (generated by Chromas 2.6.6), confirming that there is a 3bp (TGC) deletion on the der(X) chromosome and a 4bp (GGTA) insertion (indicated by the yellow box) on the der(10) chromosome.

**Figure 6 ijms-23-13076-f006:**
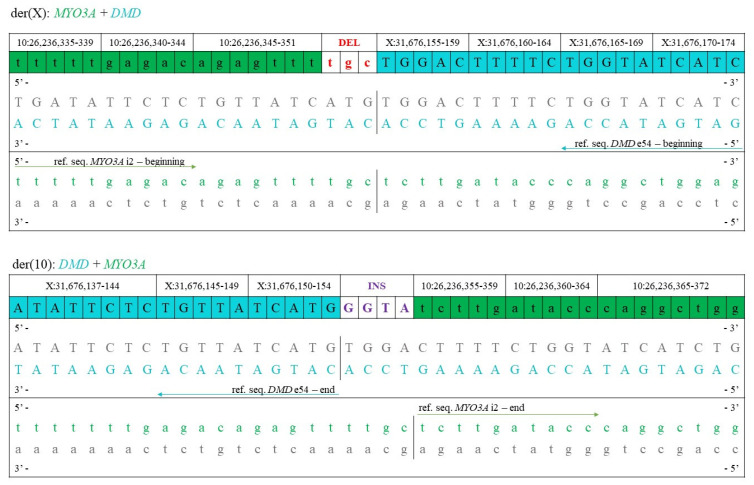
Representation of the der(X) and der(10) chromosome sequences at the breakpoints showing that there is a 3bp (TGC) deletion (DEL) on the der(X) chromosome and a 4bp (GGTA) insertion (INS) on the der(10) chromosome.

**Table 1 ijms-23-13076-t001:** Balanced translocations involving the *DMD* gene reported in the literature in females with DMD phenotype.

No.	Translocation	Additional Information	Reference
1	t(X;11)	breakpoints: Xp21 and 11q13	[18,19]
2	t(X;1)		[20]
3	t(X;5)(p21;q35)	de novo	[21]
4	t(X;6)(p21;q21)		[22]
5	t(X;9)(p21;p22)		[23]
6	t(X;3)(p21;ql3)	de novo	[23]
7	t(X;9)	de novo	[24]
8	t(X;21)(p21;p12)		[25]
9	t(X;4)(p21.1;q26)	de novo	[26,27]
10	t(X;2)(p21.2;q37)	de novo	[28]
11	t(X;5)(p21-2;q31-2)	*DMD* gene disruption in intron flanked by exons 51 and 52	[29,30]
12	t(X;15)(p21;q26)	de novo	[31]
13	t(X;4)(p21;q35)	de novo, prenatal diagnosis	[32]
14	t(X;9)(p21-2;q21-3)	de novo	[33]
15	t(X;4)(p21;q31)		[34]
16	t(X;22)	breakpoint in Xp21.2	[35]
17	t(X;12)(p21.2;q24.33)	de novo	[36]
18	t(X;7)(p21.2;p15.1)	skewed X-inactivation	[37]
19	t(X;9)(p21.1;p22.1)	array-CGH: no CNV within the *DMD* gene	[38]
20	t(X;3)(p21;p24)	de novo, prenatal diagnosis	[39]
21	t(X;4)(p21;q31)	de novo, arr(1-22,X)x2	[40]
22	t(X;4)(p21;q13)	array-CGH: no CNV within the *DMD* gene	[41]
23	t(X;13;15)(p21;q22;q22), t(6;11)(q21;q24)	WGS: *DMD* gene disruption	[42]
24	t(X;9)(p.21.1;q12)	*DMD* gene sequencing: *DMD* gene disruption	[43]
25	t(X;1)(p21.3;p22.2)	de novo	[44]
26	t(X;10)(p21.1;p12.1)	array-CGH: two VUS inherited from two parents	this study

Abbreviations: CGH—comparative genomic hybridization; CNV—copy number variation; WGS—whole genome sequencing; VUS—variant of uncertain significance.

**Table 2 ijms-23-13076-t002:** Characteristics of the t(X;10)(p21.1;p12.1) balanced reciprocal chromosomal translocation detected in the index patient. Breakpoints are shown in bold.

	der (10)	der (X)
Breakpoint in chrX	Xp21.1	Xp21.1
Coordinates of chrX	1–**31,676,155**	**31,676,158–**155,270,560
Breakpoint in chr10	10p12.1	10p12.1
Coordinates of chr10	**26,236,355–**135,534,747	1–**26,236,351**

## Data Availability

The data presented in this study are available on request from the corresponding author.

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
