# Peer review of "An Ultra-Rare Manifestation of an X-Linked Recessive Disorder: Duchenne Muscular Dystrophy in a Female Patient"

_ijms, 2022, doi:10.3390/ijms232113076_

Round 1

Reviewer 1 Report

The manuscript presents an interesting case of DMD in a female patients. The manuscript should be improved by including more recent studies. 

The text should be reevaluated, for exemple, in the introduction is mentioned "To date, more than 20 balanced reciprocal translocation cases involving the DMD gene have been reported in the literature in females, causing Duchenne muscular dystrophy phenotype" while at the Discussion is mentioned:  "To date, 25 DMD translocation cases have been published in the literature in females with DMD phenotype". Even though is not actuaally a misstake, it is better to harmonize the text.

Most of the references are older although there are published several recent studies that should be reviewed. 

Author Response

Answers to the reviewer’s comments on the manuscript entitled

“An ultra-rare manifestation of an X-linked recessive disorder: Duchenne muscular dystrophy in a female patient”

First of all, we would like to thank the reviewer for their supporting comments and suggestions in order to improve the quality of the manuscript. Here we address the comments received as follows:

“The manuscript presents an interesting case of DMD in a female patients. The manuscript should be improved by including more recent studies.

The text should be reevaluated, for exemple, in the introduction is mentioned "To date, more than 20 balanced reciprocal translocation cases involving the DMD gene have been reported in the literature in females, causing Duchenne muscular dystrophy phenotype" while at the Discussion is mentioned: "To date, 25 DMD translocation cases have been published in the literature in females with DMD phenotype". Even though is not actuaally a misstake, it is better to harmonize the text.

Most of the references are older although there are published several recent studies that should be reviewed.”

As suggested by the reviewer, “more than 20 balanced reciprocal translocation cases...” has been changed to “25 balanced reciprocal translocation cases...” in section 1.5.

The reviewer suggested that more recent studies should be included. We have found no new DMD translocation cases in female patients other than the ones described in Table 1. However, we have updated the Introduction section with the following additional information and references:

  • If given proper treatment, the majority of the patients can live until the age of 20 to 40 years, when they usually die of cardiac and/or respiratory failure (Duan et al., 2021).
  • Frameshift and nonsense mutations in the DMD gene disrupt dystrophin function, resulting in connection loss between the actin cytoskeleton and extracellular matrix. In this case, muscle fibers are easily damaged during contraction, which results in chronic muscle damage, inflammation and eventually loss of muscle function, characteristic for the DMD phenotype (Aartsma-Rus et al., 2016).
  • DMD gene mutations not disrupting the open reading frame cause a milder phenotype, Becker muscular dystrophy, the ‘reading frame rule’ explaining the different spectra of the disorder (Duan et al., 2021).
  • Defects in muscle contraction can lead to several types of diseases. The ones occurring specifically in muscle tissue proteins affect their function, thus leading to severe phenotypic presentation such as the ones mentioned above (Dowling et al., 2021).
  • The full length of the dystrophin protein is expressed in all striated skeletal, smooth and cardiac muscles, whereas shorter isoforms are expressed in brain cells and the retina (Le Rumeur et al., 2015).
  • Even in the absence of skeletal muscle symptoms, females with DMD gene variations are estimated to have a lifetime risk of developing cardiomyopathy of up to 17% (Bourke et al., 2022).
  • Further symptoms that can present include clumsiness in childhood, myalgia/cramps, unexplained abdominal or chest pain, pseudohypertrophy of the calf muscle and severe gait problems (Gruber et al., 2022).

We hope that we were able to address the issues raised by the Reviewer and would like to thank again for their work which improved our manuscript substantially.

Debrecen, 21th October 2022.

Thank you for your consideration.

Yours sincerely,

István Balogh

Division of Clinical Genetics

Department of Laboratory Medicine

University of Debrecen, Faculty of Medicine

Nagyerdei krt. 98, 4032 Debrecen, Hungary

Tel.: +36 52 340006

Fax: +36 52 417631

[email protected]

Reviewer 2 Report

In this study the authors describe a rare case of female DMD patient which results from a balanced X-autosome reciprocal translocation disrupting the DMD gene and highly skewed X-inactivation. Overall, the paper is descriptive and reports the results from various tests used to make the diagnosis of DMD for this patient. I have the following concerns/suggestions:

1. The authors carefully evaluate the breakpoints in the DMD gene and do confirmatory tests to show that it is very likely that the symptoms in their patients are because of DMD. However, there is little mention of what the effects of the disruption of MYO3A gene would be. It will be helpful to include information on the functions of MYO3A and how the breakpoints would affect its expression and/or have any clinical presentation. Please describe if the patient has any symptoms consistent with those associated with mutations in the MYO3A gene in humans.

2. Please define all abbreviation upon first use and then use them consistently. For example, DMD is used as abbreviation sometimes and in full form at other times. Please check that all abbreviations used in the manuscript have been defined.

3. Please define all abbreviations used in the Table and Figures in their respective legends.

4. Figure 1, results of muscle biopsy, please show the data from positive and negative control samples. The authors mention in line 145 on page 4 that "all immunohistochemical reactions are performed in parallel with the positive and negative controls". Please mention in the methods section the details of normal/control muscle biopsy samples. Also, please provide the catalog number of the antibodies used.

5. The results of cytogenetic testing are not shown. Please either show the results here or give reference if they have been published before.

6. Figure 3, please provide better description of the figure in the legend. Mention what is indicated by blue arrows, grey and blue circles and grey arrowheads.

7. In Figure 4, please describe what the figure shows, for example, mention which software was used to generate the traces. Please mention what the yellow highlighted area indicates.

8. Figure 5, please provide details in the legend about how the sequence data was obtained and analyzed and if any software was used to represent the data. Please define the abbreviations used in the figure (DEL and INS).

9. On page 11, line 313, please provide the source of MGIEazy Universal DNA library prep kit.

Author Response

Answers to the referees’ comments on the manuscript entitled

“An ultra-rare manifestation of an X-linked recessive disorder: Duchenne muscular dystrophy in a female patient”

First of all, we would like to thank the reviewer for their supporting comments and suggestions in order to improve the quality of the manuscript. Here we address the comments received as follows:

1) “The authors carefully evaluate the breakpoints in the DMD gene and do confirmatory tests to show that it is very likely that the symptoms in their patients are because of DMD. However, there is little mention of what the effects of the disruption ofMYO3A gene would be. It will be helpful to include information on the functions of MYO3A and how the breakpoints would affect its expression and/or have any clinical presentation. Please describe if the patient has any symptoms consistent with those associated with mutations in the MYO3A gene in humans.”

As suggested by the reviewer, we have updated the Discussion section regarding the MYO3A gene with the following:

“The other breakpoint is localised in the intron of the MYO3A gene. MYO3A encodes an actin-dependent motor protein and it is associated with autosomal recessive hearing impairment and deafness [47]. 25 disease-causing mutations have been described in the HGMD Professional 2022.3 and 26 pathogenic or likely pathogenic variants in the ClinVar (2022.10.19), the majority being loss-of function ones. However, in the gnomAD population database, there are roughly 200 alleles with loss-of-function mutations, these being present in the healthy population in heterozygous form. Moreover, our patient does not have any symptoms of hearing impairment, therefore it is highly unlikely that the disruption caused by the breakpoint causes any symptoms in this case.”

2-3) “Please define all abbreviation upon first use and then use them consistently. For example, DMD is used as abbreviation sometimes and in full form at other times. Please check that all abbreviations used in the manuscript have been defined.”

“Please define all abbreviations used in the Table and Figures in their respective legends.”

All abbreviations have been carefully reviewed and defined as requested by the reviewer.

4) “Figure 1, results of muscle biopsy, please show the data from positive and negative control samples. The authors mention in line 145 on page 4 that "all immunohistochemical reactions are performed in parallel with the positive and negative controls". Please mention in the methods section the details of normal/control muscle biopsy samples. Also, please provide the catalog number of the antibodies used.”

The methods section has been completed as follows:

“Regarding detection of sarcolemmal and sarcolemma-associated proteins, immuno-histochemistry was performed by using antibodies targeting different dystrophin epitopes: NCL-DYS1 (clone: DY4/6D3), NCL-DYS2 (clone: DY8/6C5) and NCL-DYS3 (clone: DY10/12B2), corresponding to central/core, C-terminal and N-terminal regions, respec-tively (Novocastra, Newcastle, UK, primary antibody dilution was 1:20 in all cases). The following antibodies were used targeting sarcoglycan alpha, beta, gamma, and delta: NCL-L-a-SARC, clone AD1/20A6; NCL-L-b-SARC, clone BEATASARC1/5B1; NCL-g-SARC, clone 35DAG/21B5; NCL-d-SARC, clone DELTASARC/12C1; (Novocastra, 1:50, 1:100, 1:100, and 1:50, respectively); merosin: NCL-MEROSIN, clone: MER3/22B2 (Novocastra, 1:100) and spectrin NCL-SPEC1, clone RBC2/3D5 (Novocastra, 1:100). Primary antibodies were incubated on slides for 1 hour at room temperature.

As negative control, a muscle biopsy sample from an ALS (Amyotrophic Lateral Sclerosis) patient was used. We have succesfully applied the settings described above to detect dystrophinopathy recently [49].”

Figure 2 (previously figure 1) has been completed with images from the control sample (Figure 2G-J).

5) “The results of cytogenetic testing are not shown. Please either show the results here or give reference if they have been published before.”

G-banding and FISH pattern of the t(X;10)(p21.1;p12.1) translocation has been added as Figure 1.

6) “Figure 3, please provide better description of the figure in the legend. Mention what is indicated by blue arrows, grey and blue circles and grey arrowheads.”

Completed as requested by the reviewer (Figure 3 is now Figure 4).

7)In Figure 4, please describe what the figure shows, for example, mention which software was used to generate the traces. Please mention what the yellow highlighted area indicates.”

Completed as requested by the reviewer (Figure 4 is now Figure 5), the traces were generated by Chromas 2.6.6.

8)Figure 5, please provide details in the legend about how the sequence data was obtained and analyzed and if any software was used to represent the data. Please define the abbreviations used in the figure (DEL and INS).”

Completed as requested by the reviewer (Figure 5 is now Figure 6), no special software was used to generate the figure, the sequences were copied from the Ensembl database.

9)On page 11, line 313, please provide the source of MGIEazyUniversal DNA library prep kit.”

Completed as requested by the reviewer.

We hope that we were able to answer the issues raised by the Reviewer and would like to thank again for their work which improved our manuscript substantially.

Debrecen, 21th October 2022.

Thank you for your consideration.

Yours sincerely,

István Balogh

Division of Clinical Genetics

Department of Laboratory Medicine

University of Debrecen, Faculty of Medicine

Nagyerdei krt. 98, 4032 Debrecen, Hungary

Tel.: +36 52 340006

Fax: +36 52 417631

[email protected]

Round 2

Reviewer 1 Report

The article has been significantly improved. 

Reviewer 2 Report

The authors have made the changes that I requested.